# The Prognostic and Diagnostic Value of [^18^F]FDG PET/CT in Untreated Laryngeal Carcinoma

**DOI:** 10.3390/jcm12103514

**Published:** 2023-05-17

**Authors:** Akram Al-Ibraheem, Ahmed Saad Abdlkadir, Dhuha Al-Adhami, Taher Abu Hejleh, Asem Mansour, Issa Mohamad, Malik E. Juweid, Ula Al-Rasheed, Nabeela Al-Hajaj, Dima Abu Laban, Enrique Estrada-Lobato, Omar Saraireh

**Affiliations:** 1Department of Nuclear Medicine and PET/CT, King Hussein Cancer Center (KHCC), Al-Jubeiha, Amman 11941, Jordan; 2Department of Radiology and Nuclear Medicine, Division of Nuclear Medicine, University of Jordan, Amman 11942, Jordan; 3Department of Medical Oncology, King Hussein Cancer Center, Amman 11941, Jordan; 4Department of Diagnostic Radiology, King Hussein Cancer Center, Amman 11941, Jordan; 5Department of Radiation Oncology, King Hussein Cancer Center, Amman 11941, Jordan; 6Nuclear Medicine and Diagnostic Section, Division of Human Health, International Atomic Energy Agency, 1220 Vienna, Austria; 7Department of Surgical Oncology, King Hussein Cancer Center, Amman 11941, Jordan

**Keywords:** PET/CT, semiquantitative PET parameters, laryngeal squamous-cell carcinoma, metastatic lymph node, [^18^F]FDG PET/CT vs. neck MRI, MTV, TLG

## Abstract

This study aims to determine the diagnostic accuracy of staging PET/CT and neck MRI in patients with laryngeal carcinoma and to assess the value of PET/CT in predicting progression-free survival (PFS) and overall survival (OS). Sixty-eight patients who had both modalities performed before treatment between 2014 and 2021 were included in this study. The sensitivity and specificity of PET/CT and MRI were evaluated. PET/CT had 93.8% sensitivity, 58.3% specificity, and 75% accuracy for nodal metastasis, whereas MRI had 68.8%, 61.1%, and 64.7% accuracy, respectively. At a median follow-up of 51 months, 23 patients had developed disease progression and 17 patients had died. Univariate-survival analysis revealed all utilized PET parameters as significant prognostic factors for OS and PFS (*p*-value < 0.03 each). In multivariate analysis, metabolic-tumor volume (MTV) and total lesion glycolysis (TLG) predicted better PFS (*p*-value < 0.05 each). In conclusion, PET/CT improves the accuracy of nodal staging in laryngeal carcinoma over neck MRI and adds to the prognostication of survival outcomes through the use of several PET metrics.

## 1. Introduction

Laryngeal cancer is one of the most common smoking-related head-and-neck (H&N) cancers [1]. More than 95% of laryngeal tumors are squamous-cell carcinomas [2]. Locoregional treatments of advanced non-metastatic laryngeal cancer may include total laryngectomy, ipsilateral and/or bilateral neck dissection, adjuvant-radiation therapy, or a combination of these treatments, as clinically indicated [3]. Accurate staging of laryngeal cancer is essential for selecting the most effective treatment and achieving the best possible outcome. Conventional-imaging (CI) studies, such as computed tomography (CT) and magnetic-resonance imaging (MRI), have been widely used as non-invasive imaging modalities for the detection of the primary tumor and regional nodal metastases. These imaging modalities lack the ability to distinguish between metastatic and non-metastatic cervical lymph nodes [4].

[^18^F]fluorodeoxyglucose ([^18^F]FDG) PET/CT is increasingly being used to characterize metastases at the local, regional, and distant levels [5,6]. The advantage of [^18^F]FDG PET/CT is that it combines the functional–metabolic information of PET with the detailed anatomical–morphological information of CT in a single hybrid study. PET/CT can be used to assess metabolic activity in cervical lymph nodes. Hypermetabolic disease carries a high potential for metastatic processes. Therefore, it can provide concrete evidence for determining metastatic lymph nodes. Semiquantitative PET parameters such as total lesion glycolysis (TLG) and metabolic-tumor volume (MTV) have previously shown promising results in providing a prognostic background for the disease progression of respiratory tumors [7]. Previous meta-analyses [5,6,8] have demonstrated comparable performance between [^18^F]FDG PET/CT and CI studies in the evaluation of head-and-neck cancers, including laryngeal cancer. Specific PET/CT studies on laryngeal cancer are still poorly evaluated in the literature. This requires further investigation to better understand the clinical value of this trending modality. The main purpose of this study is to determine the diagnostic accuracy of [^18^F]FDG PET/CT and neck MRI in patients with laryngeal squamous-cell carcinoma (LSCC). In addition, this study assesses the value of PET/CT parameters along with other known clinicopathologic factors in predicting progression-free survival and disease-specific survival. Finally, this study investigate the difference between PET/CT and neck MRI in achieving accurate staging results.

## 2. Materials and Methods

### 2.1. Patients

Data from a tertiary-referral cancer center were retrospectively collected and analyzed. A cohort of patients diagnosed with laryngeal cancer between December 2014 and January 2021 was investigated using the hospital information system (HIS). Among this group of patients, only 68 of them had undergone post-imaging neck dissection with biopsy-proven laryngeal malignancy (Table 1). Patients who had not had surgical resection or whose pathology results were not available were excluded from the study. Additionally, patients with synchronous tumors and patients for whom a baseline-imaging study was not available were also excluded.

All medical records of the included patients were retrospectively retrieved and analyzed to review patient, tumor, and treatment characteristics, including age at diagnosis, gender, tumor grade, tumor size, primary tumor site, T- and N-category according to the American Joint Committee on Cancer 8th edition (AJCC), tumor grade, and treatment modality [9]. PET-specific semiquantitative parameters, the pattern of enhancement on MRI, and treatment modalities were also examined. Patients were managed according to our institutional guidelines.

### 2.2. [18F]FDG PET/CT Imaging and Interpretation

[^18^F]FDG PET/CT was performed following a minimum of 4–6 h of fasting with glucose levels not exceeding 200 mg/dl. The [^18^F]FDG dose administered was between 3 and 5 MBq/kg. After injection, patients remained at rest for around 60 min in a room prepared for this purpose. A CT scan of the body was performed craniocaudally from the base of the vertex to the mid-thigh using a free-breathing signal. This procedure was carried out using a Biograph mCT PET/CT machine (Siemens Medical Solutions, Erlanger, Germany).

[^18^F]FDG PET/CT scans were interpreted visually and semi-quantitatively by two nuclear-medicine specialists with more than 10 years of work experience in [^18^F]FDG PET/CT evaluation. Maximum-intensity-projection (MIP) images were evaluated with different intensity scales, after which the images were displayed side by side. The axial, sagittal, and coronal PET reconstructions were handled with and without attenuation correction. Corresponding CT images were acquired using a 64-slice Biograph mCT flow CT and reconstructed in the axial, sagittal, and coronal planes. PET images were acquired in a 3D mode (FlowMotion technology, Erlangen, Germany), and order-subset expectation maximization (OSEM) was used to reconstruct the image. Low-dose CT without intravenous contrast was used for attenuation correction and anatomical localization. The slice thickness was 5 mm. These images were reviewed simultaneously with the PET images. The location of lesions was determined by CT scans. Patients’ weights were measured on a regular basis before IV administration of a radioisotope.

Several PET parameters were implemented to assess primary lesions, metastatic lymph nodes, and hypermetabolic lesions. Standardized uptake values (SUVs) were calculated using the contour-threshold method (VOI). The VOI was manually defined using Syngo.via (Siemens Medical Solutions Inc., Knoxville, TN, USA) [10]. One researcher manually drew a circle that was large enough to encompass all visual tumor uptake, and a trained nuclear physician reviewed it. The SUVbw map was automatically generated with software. Maximum SUV (SUVmax), metabolic-tumor volume (MTV), and total lesion glycolysis (TLG) were all calculated. The SUVmax calculations of the VOI were carried out as follows: (delayed corrected activity/tissue volume)/(injected dose/body weight). MTV is defined as the total tumor volume in voxels that is employed to estimate the total extent of activated tumor cells, based on the SUV threshold. MTV was measured using a semi-automated approach based on 40% of the maximum SUV normalized to body weight (SUVbw). A threshold of 40% of the maximum SUVbw was used to define the metabolic activity of the tumor, and the MTV was calculated using Syngo.via software [11]. The software could also render TLG values automatically by multiplying the SUVmean by the MTV of each hypermetabolic lesion. Syngo.via enabled automatic extraction of the PET parameters SUVmax, TLG, and MTV. All of the aforementioned PET parameters were examined and used as prognostic variables.

### 2.3. Neck MR Imaging and Interpretation

To carry out MR imaging of the neck, 1.5-T and 3T units were used. A 30 cm-wide head coil was used to cover the area from the frontal sinuses to the C5–C6 level. All of the patients had the axial, sagittal, and coronal T1-weighted turbo-spin echo (TSE) and axial and coronal T2-weighted m-DIXON, in addition to axial, coronal, and sagittal T1-weighted post-contrast-material m-DIXON. Axial T1-weighted FSE images were acquired with a repetition time ms/echo time ms of 670/18, an echo-train length of 5, two signals, a 240 mm field of view, a 225 × 240 matrix, a 4 mm-thick section, and a 0.4 mm gap. Axial T2-weighted m-DIXON was acquired with a repetition time m.s/echo time ms of 2500/80, an echo-train length of 19, a 240 mm field of view, a 250 × 380 matrix, a 4 mm-thick section, and a 0.4 mm gap. Axial T1-weighted post-contrast-material m-DIXON was acquired with a repetition time ms/echo time ms of 500/7, an echo-train length of 5, a 240 mm field of view, a 225 × 225 matrix, a 4 mm-thick section, and a 0.4 mm gap. For the contrast-enhanced series, a 0.2 mmol/kg of a body-weight intravenous-bolus injection of gadolinium-based contrast agent was administered at a rate of 2 mL/s. A series of baseline neck-MRI studies was performed by an experienced radiologist. These images were performed no more than two weeks apart from or within the same timeframe of [^18^F]FDG PET/CT-image acquisition. Cervical lymph nodes were considered metastatic if central necrosis or a heterogeneous enhancement pattern was present. Diameter cutoff values of more than 1.5 cm in the jugulodigastric and submandibular regions and those exceeding 1 cm in all other cervical-lymph-node levels were also considered metastatic. Additionally, the metastatic process was considered when the ratio of the maximum longitudinal nodal length to the maximum axial nodal length (L/T) was less than 2, or if there was a group of three or more lymph nodes with borderline size. Regional-lymph-node groups were subdivided into six levels for each side using the locations and anatomic boundaries of cervical-lymph-node groups.

### 2.4. Reference Standard

In order to determine [^18^F]FDG PET/CT and neck-MRI accuracy, a group of histopathological criteria was collected and retrieved from post-imaging biopsy results. The results of primary-tumor and cervical-nodal involvement were checked and compared with the imaging results to determine accuracy. 

The Radiology Information System (RIS) was used to obtain the initial interpretation of the imaging studies, which included PET/CT scans and MRIs. A random sample of cases was chosen in order to assess the accuracy and completeness of the data. The corresponding imaging data of PET/CT and MRI reports were then reviewed by a trained nuclear-medicine specialist and radiologist. The specialists who reviewed the cases were blinded to any clinical or pathological information about the patients. They checked for discrepancies or errors in the radiology reports and imaging data. Any discrepancies or errors were recorded and accounted for in the data analysis. Following that, all PET/CT scans were reviewed by a trained nuclear-medicine specialist who was not given any information about the patients’ clinical and pathological data. The specialist extracted the PET/CT scan’s various parameters without impacting the interpretation criteria. The parameters of interest were SUVmax, MTV, and TLG. These parameters were then used in the analysis to see how they related to the pathological findings and clinical outcomes. 

In comparison to histologic staging, these results were classified as concordant or discordant. If they agreed with the histologic staging (agreement of the initial MRI and PET/CT reports or by reaching a consensus), they were regarded as concordant. Concordance with biopsy findings can be achieved if the overall nodal-staging category is similar to the biopsy findings, regardless of the level or number of involved lymph nodes. In contrast, when findings of one or both imaging modalities disagreed with histologic staging, it was considered discordant. In some circumstances, this discordance may be either mismatch (only single-modality discordance) or bimodality discordance.

### 2.5. Statistical Analysis

Sensitivity, specificity, and overall accuracy were calculated for each diagnostic test, along with 95% confidence intervals. The accuracy of combined [^18^F]FDG PET/CT and MR imaging was also estimated with a 95% confidence interval. The McNemar test was used to compare data obtained separately from [^18^F]FDG PET/CT and neck MRI. Furthermore, the same statistical test was used to compare data from the two methods used together versus data from either modality considered separately. Any *p*-values less than 0.05 were considered statistically significant.

The Kaplan–Meier method was used to plot survival curves for progression-free survival (PFS) and overall survival (OS). The log-rank test was used to assess survival differences between subgroups. ROC-curve analysis was employed to determine optimal cutoff values for each continuous variable (e.g., semiquantitative PET parameters).

The Cox proportional-hazards regression model was used in both univariate and multivariate analyses to assess the relationship between survival, clinicopathological factors, and several semiquantitative PET parameters. Any variable in the univariate model with a *p*-value greater than 0.05 was eligible for multivariable Cox modeling. In the multivariate analysis, a significant threshold of a *p*-value greater than 0.05 was established. SPSS 26.0 for Windows (SPSS, Inc., Chicago, IL, USA) was used for all analyses.

## 3. Results

### 3.1. Patient and Tumor Characteristics

A total of 68 patients with LSCC had staging PET/CT and MRI performed before treatment; 67 (98.5%) were male and only 1 (1.5%) was female. Average tumor size of the primary lesion was 3.8 cm (range of 1.5–7.5 cm). Primary-tumor sites were found within the glottis (39 patients), supraglottis (26 patients), and subglottis (3 patients) (Table 1). The majority of cases had a moderately differentiated tumor (about 63% of cases). Patients were staged according to the 8th edition of the AJCC system [9] as follows: stage III (*n* = 6), stage IVA (*n* = 54), stage IVB (*n* = 7), and stage IVC (*n* = 1) (Table 1). Histopathologic results reported cervical-lymph-node metastasis due to laryngeal cancer in 32 patients.

### 3.2. Primary Tumor

All primary tumors were clearly identified by both [^18^F]FDG PET/CT and neck MRI. Primary-tumor lesions tend to attain a high level of metabolic activity, which is reflected by high values of different PET parameters. In all cases examined, [^18^F]FDG uptake values were noticeably high, with an average value of 16.1 for SUVmax, 13.4 for MTV, and 150.6 for TLG. A representative case is addressed to highlight these findings (Figure 1).

### 3.3. Cervical-Lymph-Node Metastases

#### 3.3.1. Patient-Based Analysis

Among all 32 patients with histopathology-proven nodal disease, [^18^F]FDG PET/CT outperformed neck MRI, rendering more sensitive results while remaining edge to edge with MRI in terms of specificity. Sensitivity, specificity, and accuracy for nodal disease were 93.8%, 58.3%, and 75%, respectively, for [^18^F]FDG PET/CT, and 68.8%, 61.1%, and 64.7%, respectively, for neck MRI (Table 2). These values appeared to be statistically significant (McNemar test was significant with a *p*-value of 0.022).

#### 3.3.2. Side-by-Side Analysis

Post-imaging unilateral or bilateral neck dissection was performed in all cases. Among all 64 patients with previous right-side neck dissection, [^18^F]FDG PET/CT was able to detect all but one instance of biopsy-proven metastatic nodal disease. These results are reflected by the 95.7% sensitivity, 58.5% specificity, and accuracy of about 72%. Neck MRI rendered similarly accurate results, achieving an accuracy of about 69%, with a sensitivity and specificity of about 70% and 68%, respectively. Left-side neck dissection was performed in 55 cases. Patients with a history of left-side neck dissection were assessed in a similar fashion. [^18^F]FDG PET/CT scored about 95.5% and 60.6% for sensitivity and specificity, respectively, with an overall accuracy of about 75%. On the other hand, neck MRI achieved about 73%, 55%, and 62% for sensitivity, specificity, and accuracy, respectively (Table 2).

#### 3.3.3. Assessment of Correct Nodal Staging

Compared to MRI, [^18^F]FDG PET/CT outperformed MRI in achieving correct nodal staging (Figure 2). 

Among all patients, [^18^F]FDG PET/CT achieved correct nodal staging in 45 patients, whereas MRI achieved correct staging in 24 patients. In terms of histopathology, there were 18 cases upstaged by PET/CT compared to 27 cases in MRI, whereas understaging occurred in 5 cases in PET/CT compared to 17 cases in MRI (Figure 3).

### 3.4. Role of [^18^F]FDG-PET/CT in Predicting PFS and OS

At a median follow-up period of 51 months (3–88 months), disease events were noted in 23 patients and death was observed in 17 patients. A total of 18 patients out of 23 patients who had eventful disease on follow-up experienced metastatic disease. Lung metastasis was the most prevalent pattern of relapse, followed by liver, bone, brain, and skin metastasis. Local recurrence was observed in 13 cases and was detected by MRI. Nodal-disease recurrence was witnessed in 10 out of 13 patients with recurrent disease. The median progression-free survival was 31 months (1–87 months) and the median OS was 48 months (6–88 months). In univariate analysis, considering cutoff values retrieved from receiver-operating-characteristic (ROC) curves (Table 3), all utilized semiquantitative PET parameters (i.e., SUVmax, TLG, and MTV) were significantly correlated with PFS and OS (*p* < 0.01 for each). These parameters were calculated and examined for patients during assessment of baseline [^18^F]FDG PET/CT imaging.

Tumor SUVmax, tumor MTV, and tumor TLG were also significantly correlated with OS and PFS (*p* < 0.01 each) (Figure 4). 

Using Cox regression, univariate analysis revealed all utilized PET parameters as significant prognostic factors in PFS and OS (Table 4). Finally, MTV and TLG were found via multivariate analysis to be significant in terms of predicting PFS (Table 4).

## 4. Discussion

To our knowledge, this is one of the few studies addressing the prognostic and diagnostic utility of [^18^F]FDG PET/CT in laryngeal cancer. Many of the previously published papers included all sites of head-and-neck cancers. There is only limited evidence in the literature regarding the role of molecular imaging specifically for laryngeal tumors [12,13]. Nevertheless, recent evidence has acknowledged the evolving role of PET/CT in detecting laryngeal tumors. Our results demonstrate the superior diagnostic capabilities of [^18^F]FDG PET/CT over MRI for nodal disease staging. [^18^F]FDG PET/CT was more accurate than neck MRI in 21 out of 68 cases. The [^18^F]FDG PET/CT scan performed well in terms of sensitivity (95.5%) and specificity (60.6%), with an overall accuracy of 75%. The neck MRI scan, on the other hand, scored lower in terms of sensitivity (68.8%), specificity (61.1%), and accuracy (64.7%). Additionally, semiquantitative PET metrics implemented from the primary tumor in patient-based analysis have shown a significant impact on survival outcomes. All of the semiquantitative PET parameters that were used (i.e., SUVmax, TLG, and MTV) were significantly correlated with PFS and OS. Additionally, MTV and TLG were significant predictors of PFS using univariate and multivariate survival.

Currently, CT and MRI of the head and neck are used to detect local disease, tumor burden, and nodal involvement. MRI has better soft-tissue contrast resolution than CT and is more sensitive and specific in determining local disease extension [7]. Similarly, [^18^F]FDG PET/CT is becoming more important in staging and assessing treatment response in patients with laryngeal cancer [14]. However, due to the partial-volume effect and limited resolution, [^18^F]FDG PET/CT may result in underestimation of primary tumors at small sizes [14]. Brown-fat uptake, in addition to muscular and vocal-cord uptake, are common confounders during local-disease assessment [14]. Therefore, neck MRI is the primary modality of choice for detecting primary-tumor lesions. Despite this, recent studies have observed certain scenarios wherein [^18^F]FDG PET/CT can be more powerful in terms of accuracy than neck MRI. A study of 35 patients by Chaput et al. showed that [^18^F]FDG PET/CT had greater diagnostic accuracy, with a sensitivity of 83%, compared to MRI, which was only 63% sensitive for detecting head-and-neck cancers in patients in the T1–T2 category [15]. This suggests that [^18^F]FDG PET/CT is more effective in lower-stage disease. When PET is performed with a contrast-enhanced CT, diagnostic CT images can be obtained from a single examination, and this may eliminate the need for additional imaging for staging purposes (CT or MR) [16].

Accurate assessment of nodal disease during the initial workup is essential in order to predict the likelihood of distant metastases and achieve successful local control of the disease following treatment [17]. Both neck MRI and [^18^F]FDG PET/CT are well-established methods for detecting nodal disease. Neck MRI is limited to the morphologic evaluation of detectable lymph nodes, such as size, margins, and necrosis. Thus, it is rather challenging to determine nodal disease using MRI alone. This is especially true for patients who have non-enlarged, suspicious lymph nodes [18]. One of the previous meta-analyses demonstrated that [^18^F]FDG PET/CT can improve nodal detectability by 5–10% [8]. Overall sensitivity and specificity values observed with [^18^F]FDG PET/CT ranged between 86 and 98% for both [19,20,21]. [^18^F]FDG PET/CT can provide additional metabolic-background features to assist in detecting suspicious lymph nodes. This can be achieved with the help of semiquantitative PET parameters. A meta-analysis of 724 head-and-neck-cancer patients found that PET/CT could accurately detect regional nodal metastases [5]. The overall observed sensitivity and specificity were 84 and 84–96%, respectively [5]. Therefore, [^18^F]FDG PET/CT can alter treatment in tumors like supraglottic laryngeal carcinoma, which often involves bilateral lymph nodes [22]. 

By far, [^18^F]FDG PET/CT is known to be the modality of choice for distant metastatic M staging. Krabbe et al. reported high sensitivity and specificity values (exceeding 90%) for detecting distant disease using [^18^F]FDG PET/CT [23]. In contrast, CI demonstrated only moderate degrees of both sensitivity and specificity [23]. More recently, Rohde et al. showed that PET/CT is better than a combination of radiographs, CT, and MRI for detecting distant metastases and second primary malignancies [24]. Usual sites of metastases are frequently observed to involve the lungs, liver, and bones [25]. Adding this benefit of [^18^F]FDG PET/CT to our findings about superiority in nodal staging and the added prognostic value of semiquantitative parameters results in a superior modality for laryngeal-cancer staging.

The role of [^18^F]FDG PET/CT in disease prognostication has been increasingly recognized as important [26]. This is because disease recurrence and/or progression are common, and early assessment of therapy response and prospective-survival analysis can help optimize treatment plans for advanced cases. The prognostic value of [^18^F]FDG-PET/CT semiquantitative parameters has been demonstrated in several neoplastic diseases, including laryngeal cancer [27,28,29]. The SUVmax measurement parameter is the most well-known and widely reported semi-quantitative PET measurement. SUVmax has been identified as a prognostic factor in head-and-neck squamous-cell carcinoma [30]. High levels of SUVmax are frequently associated with poor disease prognosis. However, more recent meta-analyses have shown that MTV and TLG are more accurate in predicting prognosis than SUVmax [31]. It should be noted that TLG and MTV are not yet widely adopted. This is mainly due to the difficulties encountered and the requirement of specialized software [32].

In our univariate analysis, all of the utilized semiquantitative parameters showed a significant correlation with survival outcomes. However, in our multivariate study, only MTV and TLG were found to be significant predictors of PFS. This evidence is consistent with previous similar studies [29,32,33]. In fact, high tumor TLG and MTV are known to represent a great degree of tumor aggressiveness, which impacts the survival outcome [30,34]. However, these parameters are not yet widely established despite their great impact, as judged by previous studies. The current evidence from the literature suggests that there are many benefits to be gained from using these parameters. This can help to confirm existing evidence and may establish a new roadmap for management that can enhance survival outcomes.

A recent trend has been the use of machine learning and artificial intelligence in cancer diagnosis and prognosis [34]. These can provide more quantitative data to be used during the initial workup and for staging purposes [34]. [^18^F]FDG PET/CT is one example of this and is being investigated for use in head-and-neck cancers [35,36]. This can minimize acquisition time, reduce costs, and improve patients’ experience. However, more research is needed to help strengthen the diagnostic and prognostic utility of [^18^F]FDG PET/CT. Adopting contemporary imaging methods such as diffusion-weighted imaging (DWI) can be advantageous. In fact, DWI is known to reduce false findings and increase overall specificity [37]. Artificial intelligence can be incorporated into MRI to speed up acquisition time.

This study has several limitations, including its retrospective nature, single-center experience, and relatively small number of patients analyzed. Finally, due to the nature of the [^18^F]FDG PET/CT modality, there is a known limitation related to false-positive inflammatory lymph nodes, which were observed in a minority of patients. However, it is one of the few studies arguing for the diagnostic and prognostic role of [^18^F]FDG PET/CT specifically for laryngeal carcinoma.

## 5. Conclusions

In laryngeal cancer, [^18^F]FDG PET/CT is superior to neck MRI in terms of achieving higher accuracy in detecting nodal metastasis. Additionally, the semiquantitative parameters of PET were found to be independent prognostic factors that correlate with survival in the same patient population. These parameters could further stratify the prognosis of these patients, and these findings might provide solid functional-imaging evidence for future studies to pursue.

## Figures and Tables

**Figure 1 jcm-12-03514-f001:**
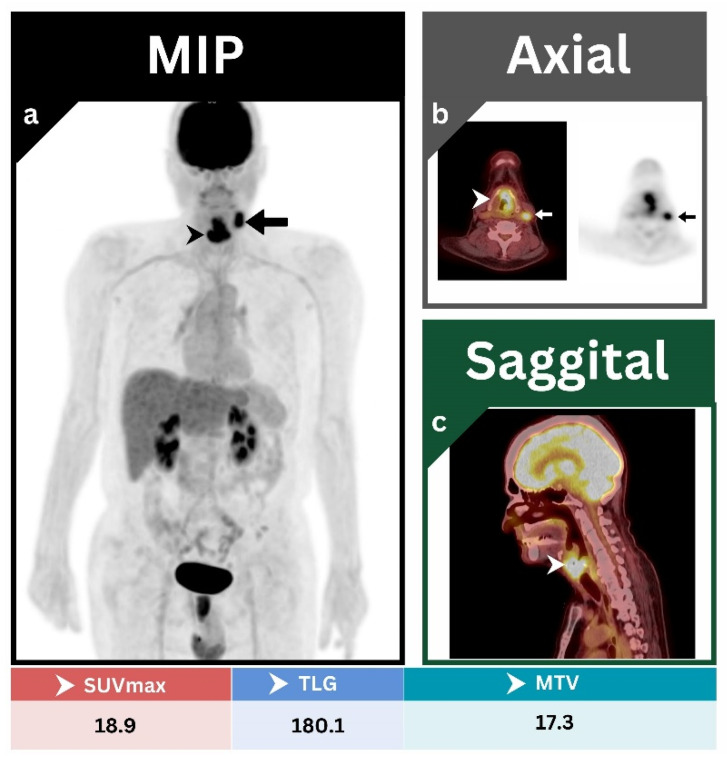
(**a**) MIP; (**b**) axial PET/CT, and axial PET; (**c**) sagittal PET/CT images showing an intensely hypermetabolic malignant glottic-mass lesion (arrowheads) with direct invasion to left para-glottic space and thyroid cartilage. This was associated with a single hypermetabolic left level III cervical lymph node (arrows). The overall staging result was T4aN1M0, which is consistent with the post-operative biopsy.

**Figure 2 jcm-12-03514-f002:**
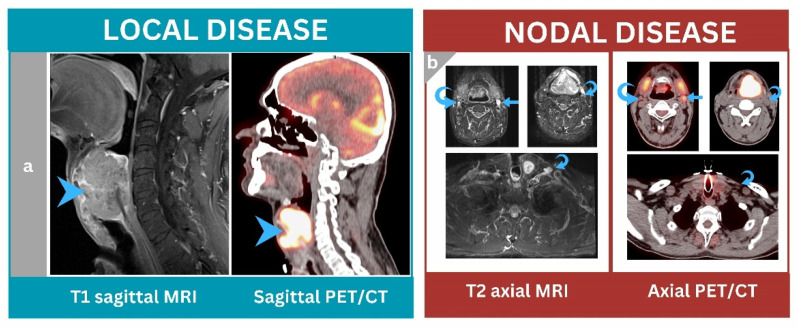
A 49-year-old male patient underwent baseline neck MRI followed by [^18^F]FDG PET/CT for staging purposes. (**a**) Sagittal PET/CT and sagittal T1 MRI images showed a large lobulated transglottic-mass lesion with an extra-laryngeal extension that was highly [^18^F]FDG avid and heterogeneous on MRI (arrowheads). (**b**) Axial PET/CT and axial T2 MR images showed a prominent left level IIa cervical lymph node that was hypermetabolic and enhanced by MRI (arrows), suggesting metastases. Other right level IIa and left III–IV lymph nodes were heterogeneously enhanced but non-[^18^F]FDG avid (curved arrows). [^18^F]FDG PET/CT nodal staging outperformed MRI and matched the biopsy results (N1 nodal disease).

**Figure 3 jcm-12-03514-f003:**
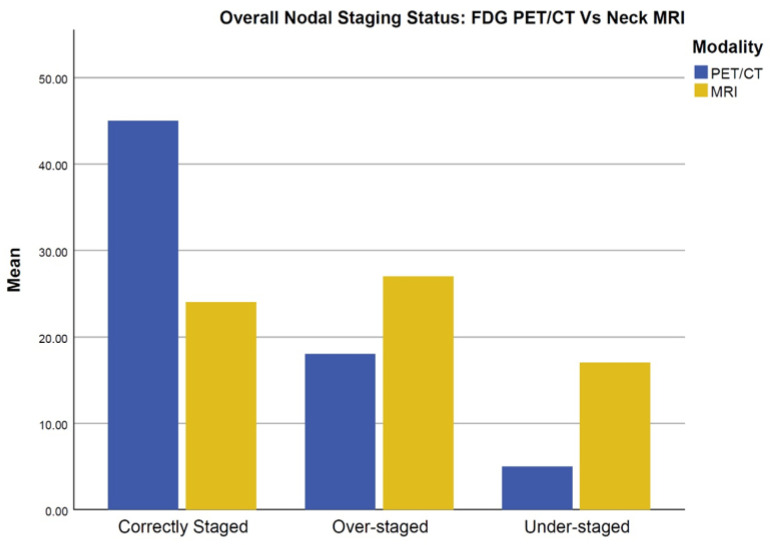
Overall nodal-staging status: [^18^F]FDG PET/CT vs. neck MRI.

**Figure 4 jcm-12-03514-f004:**
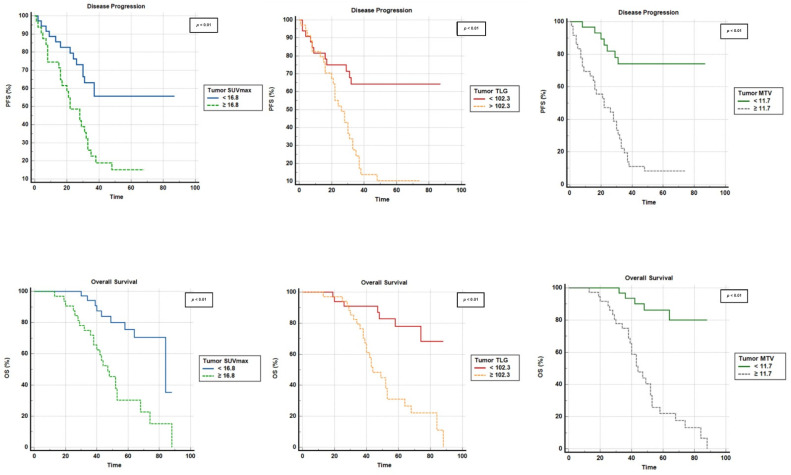
Kaplan–Meir survival curves according to baseline tumor SUVmax, TLG, and MTV.

**Table 1 jcm-12-03514-t001:** Demographic, histopathological, and clinical characteristics of study sample.

Demographics	
Age (in Years)	
Median	58.5 years
Range	45–85 years
Gender (Number, Percentage)	
Male	67, 98.5%
Female	1, 1.5%
Histopathological Characteristics	
Tumor Site (Number, Percentage)	
Glottic	39, 57.4%
Supraglottic	26, 38.2%
Subglottic	3, 4.4%
Tumor Size (Cm Mean & Range)	3.8 cm, (1.5–7.5 cm)
T-Category (Number, Percentage)	
T1	0
T2	2, 3%
T3	10, 15%
T4	56, 82%
N-Category (Number, Percentage)	
N0	37, 55%
N1	13, 19%
N2	11, 16%
N3	7, 10%
Tumor Grade (Number, Percentage)	
Well differentiated	4, 6%
Well–moderately differentiated	1, 1.5%
Moderately differentiated	43, 63.2%
Moderate–poorly differentiated	10, 14.7%
Poorly differentiated	10, 14.7%
Treatment Modality (Number, Percentage)	
Total laryngectomy	68, 100%
Bilateral neck dissection	51, 75%
Unilateral neck dissection	17, 25%
Adjuvant chemoradiation	11, 16%
Adjuvant local radiotherapy	9, 13%

**Table 2 jcm-12-03514-t002:** Results of nodal detectability by [^18^F]FDG PET/CT and MRI studies of the neck.

Neck-Dissection Side	[^18^F]FDG PET/CT	Neck MRI
Right	-TP ^1^: 22	-TP: 16
	-FN ^2^: 1	-FN: 7
	-TN ^3^: 24	-TN: 28
	-FP ^4^: 17	-FP: 13
	-Sensitivity: 95.7%	-Sensitivity: 69.6%
	-Specificity: 58.5%	-Specificity: 68.3%
	-PPV ^5^: 56.4%	-PPV: 55.2%
	-NPV ^6^: 96%	-NPV: 80%
	-Accuracy: 71.9%	-Accuracy: 68.8%
Left	-TP: 21	-TP: 16
	-FN: 1	-FN: 6
	-TN: 20	-TN: 18
	-FP: 13	-FP: 15
	-Sensitivity: 95.5%	-Sensitivity: 72.7%
	-Specificity: 60.6%	-Specificity: 54.5%
	-PPV: 61.8%	-PPV: 51.6%
	-NPV: 95.2%	-NPV: 75%
	-Accuracy: 74.5%	-Accuracy: 61.8%
Overall	-TP: 30	-TP: 22
	-FN: 2	-FN: 10
	-TN: 21	-TN: 22
	-FP: 15	-FP: 14
	-Sensitivity: 93.8%	-Sensitivity: 68.8%
	-Specificity: 58.3%	-Specificity: 61.1%
	-PPV: 66.7%	-PPV: 61.1%
	-NPV: 91.3%	-NPV: 68.8%
	-Accuracy: 75%	-Accuracy: 64.7%
McNemar Test 0.022		

^1^ TP: true positive; ^2^ TN: true negative; ^3^ FP: false positive; ^4^ FN: false negative; ^5^ PPV: positive predictive value; ^6^ NPV: negative predictive value.

**Table 3 jcm-12-03514-t003:** Semiquantitative PET/CT parameter cutoffs calculated using ROC-curve analysis for primary laryngeal cancer during initial staging.

Parameter	ROC Curve for PFS				
	Cutoff	AUC ^1^ (95% CI)	*p*-Value	Sensitivity	Specificity
SUVmax	16.8	0.656 (0.532–0.767)	0.029	65.2%	71.7%
TLG	102.3	0.707 (0.585–0.810)	0.001	82.6%	69.6%
MTV	11.7	0.827 (0.717–0.907)	<0.001	91.3%	71.7%
	**ROC Curve for OS**				
	**Cutoff**	**AUC (95% CI)**	***p*-Value**	**Sensitivity**	**Specificity**
SUVmax	16.8	0.637 (0.512–0.749)	0.105	70.6%	69.2%
TLG	102.3	0.683 (0.560–0.790)	0.009	88.2%	65.4%
MTV	11.7	0.767 (0.650–0.860)	<0.001	88.2%	65.4%

^1^ AUC: area under curve.

**Table 4 jcm-12-03514-t004:** Univariate and multivariate survival analyses for PFS and OS.

Criterion	Univariate Analysis		Multivariate Analysis	
	*p*-Value	HR ^1^ (95% CI)	*p* Value	HR ^1^ (95% CI)
PFS				
Age	0.437	1.277 (0.689–2.366)		
Tumor site	0.883	0.972 (0.662–1.425)		
Tumor grade	0.071	0.674 (0.439–1.034)		
Smoking	0.282	0.775 (0.488–1.233)		
SUVmax	0.003	3.640 (1.530–8.662)	0.734	1.182 (0.452–3.093)
TLG	0.001	8.049 (2.704–23.961)	0.021	4.241 (1.240–14.502)
MTV	<0.001	11.994 (3.536–40.682)	0.002	7.504 (2.107–26.720)
OS				
Age	0.315	1.454 (0.701–3.019)		
Tumor site	0.038	0.551 (0.315–0.966)	0.079	0.571 (0.305–1.068)
Tumor grade	0.109	0.677 (0.420–1.091)		
Smoking	0.041	0.536 (0.295–0.975)	0.742	0.901 (0.484–1.068)
SUVmax	0.008	4.275 (1.473–12.409)	0.820	1.162 (0.318–4.246)
TLG	0.002	10.539 (2.394–46.398)	0.070	5.009 (0.874–28.694)
MTV	0.003	9.337 (2.132–40.881)	0.117	3.716 (0.718–19.228)

^1^ HR: hazard ratio.

## Data Availability

The data presented in this study are available on request from the corresponding author. The data are not publicly available due to privacy.

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
