# Peer review of "The Prognostic and Diagnostic Value of [18F]FDG PET/CT in Untreated Laryngeal Carcinoma"

_jcm, 2023, doi:10.3390/jcm12103514_

Round 1

Reviewer 1 Report

This is a well-organised study with clear presentation.

Author Response

Letter to Reviewer 1
This is a well-organized study with a clear presentation.
Thank you for your time and dedication to reviewing our work. It was very inspiring to recommend our work. Upon receiving your review, we became aware of certain linguistic imperfections and typographical errors that had eluded our attention in the earlier version. Your insightful feedback has been immensely helpful in improving the quality of our manuscript. We have also thoroughly revised the manuscript to correct any typos or errors that were present in the earlier version. Thanks to your guidance, we have achieved a more coherent and uniform style in presenting the vital points of our research. Kindly track the applied change in the revised version to account for previous errors:
• In line 28: and add was changed to and adds.
• In line 34: smoking related was changed to smoking-related.
• In line 140 and line 143: dot was inserted instead of coma to separate the two sentences
• In line 177: “P values of less than 0.05 was considered” was changed to “P values of less than 0.05 were considered”
• In lines 327-328: “which often involve” was changed to “which often involves”
• In line 351: “survival outcome” was changed to “survival outcomes”
• In lines 377-378: “higher accuracy of detecting nodal metastasis” was changed to “higher accuracy in detecting nodal metastasis”

Reviewer 2 Report

Thank you very much for the opportunity to review the manuscript entitled “The Prognostic and diagnostic Value of FDG PET/CT in un-treated laryngeal carcinoma”. The subject raised by the authors is important in the field of laryngeal cancer patients, however some changes are required before consideration for publication.

1.       Introduction section, page 2, line 51: MTV is metabolic tumor volume, not “mean tumor volume”.

2.       Material and methods section, table 1. – Please describe based on which modality tumor size was assessed.

3.       Material and Methods section, point 2.2 – please add the information in the first paragraph about slice thickness in CT and PET imaging. Moreover, please add also information about time for bed position for PET imaging.

4.       Material and Methods, point 2.3, lines 157-160: the used abbreviations for PET parameters were already explained earlier (in the point 2.2) thus there is no need to explain them once again.

5.       Material and Methods section, point 2.4 – there is no explanation of OS in PFS earlier in the main document – please explain abbreviation when they first are used

6.       Material and Methods, point 2.4 – the last paragraph is a repeating of what was already written in the point 2.3.

7.       Results section, figure 1 – there is no sense to show on the figure the results which are clearly presented in the table – I recommend to delete the figure 1. The same is referred to Figure 5 – there is no need to repeat in Figures the results which are already presented in the main text.

8.       Results section: authors should add an information (for example in the table) about values for all FDG PET assessed parameters (mean±SD) for primary tumor as well as for metastatic lymph nodes. Additionally, they should be presented in terms of stage of the disease.

9.       Results section, page 9, line 268: TLG and MTV are not a metabolic parameters – they are known as a volumetric parameters. If authors would not like to differentiate between metabolic (SUV) and volumetric parameters (TLG and MTV), they might use the term “semiquantitative PET parameters”.

10.   Discussion section: Figures should not be placed in the discussion section.

11.   Please check the whole manuscript for typos or other linguistic errors: for example (but not limited to), material and methods section, point 2.3, line 136, 139 – it looks like the dots are missing between words.

12.   Please provide an appropriate nomenclature according to "Consensus nomenclature rules for radiopharmaceutical chemistry — setting the record straight”.

Author Response

Letter to Reviewer 2
Thank you very much for the opportunity to review the manuscript entitled “The Prognostic and diagnostic Value of FDG PET/CT in un-treated laryngeal carcinoma”. The subject raised by the authors is important in the field of laryngeal cancer patients, however some changes are required before consideration for publication.
We would like to thank you very much for taking the time to review our manuscript and to provide this insightful review, which has helped us improve the quality of our manuscript.
Your respectful review points are addressed in details below.
1. Introduction section, page 2, line 51: MTV is metabolic tumor volume, not “mean tumor volume”.
Thank you for pointing out this typo. We have corrected this mistake and rechecked the manuscript for any further errors. The previous typo error is now corrected. Changes were reflected (lines 51-52) in the revised version of the manuscript.
2. Material and methods section, table 1. – Please describe based on which modality tumor size was assessed.
Thank you for bringing up this important point to our attention. All tumor characteristics involving (tumor site, tumor size, TNM, and tumor grade) have been obtained and presented in table 1 from histopathologic report. Kindly track changes applied in table 1 title to account for previous error in line 74. We have added the word (histopathological) to indicate that tumor characteristics were retrieved from biopsy results. We have also updated the table layout to further clarify the presented point of view. Kindly track changes in row 1, row 8, and row 30 within table 1 in the revised version of the manuscript.
3. Material and Methods section, point 2.2 – please add the information in the first paragraph about slice thickness in CT and PET imaging. Moreover, please add also information about time for bed position for PET imaging.
Slice thickness for both CT and PET along with time per bed position was added in lines 96-99. Additional information about PET/CT methodology was also added in the same segment.
4. Material and Methods, point 2.3, lines 157-160: the used abbreviations for PET parameters were already explained earlier (in the point 2.2) thus there is no need to explain them once again.
Duplicate abbreviations were corrected and removed from line 162.
5. Material and Methods section, point 2.4 – there is no explanation of OS in PFS earlier in the main document – please explain abbreviation when they first are used
Thank you for pointing out this missing. The explanation for both the OS and PFS abbreviations has been added at their first appearance in the main text. These changes have been reflected in lines 180-181 in the revised version of the manuscript.
6. Material and Methods, point 2.4 – the last paragraph is a repeating of what was already written in the point 2.3.
Thank you for bringing this to our attention. The previously duplicated statements were removed (from statistical Analysis segment), in the revised version of the manuscript.
7. Results section, figure 1 – there is no sense to show on the figure the results which are clearly presented in the table – I recommend to delete the figure 1. The same is referred to Figure 5 – there is no need to repeat in Figures the results which are already presented in the main text.
Previous Figure 1 and 5 have been removed in response to your input.
8. Results section: authors should add an information (for example in the table) about values for all FDG PET assessed parameters (mean±SD) for primary tumor as well as for metastatic lymph nodes. Additionally, they should be presented in terms of stage of the disease.
We appreciate your suggestion to include this information. However, I'd like to emphasize that in our study, we used a patient-based analysis method rather than a lesion-based analysis method. We believe that, within the scope of our study, creating a table with this level of detail for each patient would be not feasible and could potentially confuse our approach to highlighting FDG PET's overall performance in disease staging. We appreciate your feedback, but we hope that our explanation clarifies our viewpoint.
9. Results section, page 9, line 268: TLG and MTV are not metabolic parameters – they are known as volumetric parameters. If authors would not like to differentiate between metabolic (SUV) and volumetric parameters (TLG and MTV), they might use the term “semiquantitative PET parameters”.
The previous classification misconception was corrected replacing all metabolic PET parameters with “Semiquantitative PET parameters”. Kindly track changes in lines 30, 51, 79, 183, 186, 264, 268, 292, 294, 325, 338, 344, 353, and 381.
10. Discussion section: Figures should not be placed in the discussion section.
In response to your respectful review point, Figure attached in discussion section was removed to avoid overexpression and adhere to journal guidelines.
11. Please check the whole manuscript for typos or other linguistic errors: for example (but not limited to), material and methods section, point 2.3, line 136, 139 – it looks like the dots are missing between words.
Language quality assessment and grammar correction was done to improve the quality of the updated manuscript. Some examples (among many others) are shared below:
• In line 28: and add was changed to and adds.
• In line 34: smoking related was changed to smoking-related.
• In line 140 and line 143: dot was inserted instead of coma to separate the two sentences
• In line 177: “P values of less than 0.05 was considered” was changed to “P values of less than 0.05 were considered”
• In lines 327-328: “which often involve” was changed to “which often involves”
• In line 351: “survival outcome” was changed to “survival outcomes”
• In lines 377-378: “higher accuracy of detecting nodal metastasis” was changed to “higher accuracy in detecting nodal metastasis”
12. Please provide an appropriate nomenclature according to "Consensus nomenclature rules for radiopharmaceutical chemistry — setting the record straight”.
On this point, we altered the definition and abbreviation for [18F]fluorodeoxyglucose ([18F]FDG) per requirement.
In the end, we would like to express our sincere gratitude to the reviewer for his valuable feedback. His insightful comments have helped us to improve this manuscript and strengthen the point of view we aim to convey. By applying the required modifications, we have achieved a more coherent and uniform style in presenting the vital points of our research. We truly appreciate his time and effort in providing such constructive feedback.

Reviewer 3 Report

Dear author,

Great work for a very important points which has to be clear.

But you need to clarify in the section patient and method that its a retrospective study ( you have mentioned it only in the limitation of the study

the second points in the limitation if the FDG PET/CT has affected by Diabetus  or by any Inflammatory process

Author Response

Letter to Reviewer 3
• Great work for a very important points which has to be clear.
Thank you for your encouraging and insightful remarks. It is very encouraging to know that our efforts are being recognized and will hopefully have a positive impact on such an important topic.
Kindly note that we have chosen to subdivide each question with number labeling to facilitate the review process and answer each question accordingly.
Below are the answers to your respectful review points.
1. You need to clarify in the section patient and method that it’s a retrospective study (you have mentioned it only in the limitation of the study)
We appreciate you bringing this up. The retrospective nature of the study has been introduced and reinforced in the recent version. Kindly track changes in lines (66, and 75 in patients section) of the revised version of the manuscript.
2. In the limitation Section: You should indicate if the FDG PET/CT has been affected by Diabetes or by any Inflammatory process
Thank you for bringing this important point to our attention. It is important to note that all diabetic patients undergoing FDG PET/CT were under strict control before the imaging procedure. Additionally, any patient with FBS < 200 mg/dl will be postponed until further glycemic control is achieved (which is mentioned in lines 82-85). Some of these patients may exhibit diffuse colonic uptake due to metformin consumption, but in our cases, inflammatory conditions at the site of the primary tumor (the larynx) were not encountered. However, we do acknowledge that during nodal disease assessment, such limitations due to inflammation were encountered, and we have added this information to the limitation sections in the discussion subheading (please track changes in lines 374-376). This limitation is more pronounced when assessing nodal disease through lesion-based analysis. However, this limitation has only a trivial effect on the overall staging results obtained from patient-based analysis.

Round 2

Reviewer 2 Report

In material and method section there are two points 2.3 - please clarify this.

Author Response

Thank you for bringing this point to our attention. We have updated the number labeling of "Material and Method" in the correct ascending fashion. The previous number mislabeling was corrected thanks to your respectful review.
